# Understanding-in-Generation: Reinforcing Generative Capability of Unified Model via Infusing Understanding into Generation

## Abstract

Recent works have made notable advancements in enhancing unified models for text-to-image generation through the Chain-of-Thought (CoT). However, these reasoning methods separate the processes of understanding and generation, which limits their ability to guide the reasoning of unified models in addressing the deficiencies of their generative capabilities. To this end, we propose a novel reasoning framework for unified models, **Understanding-in-Generation (UiG)**, which *harnesses the robust understanding capabilities of unified models to reinforce their performance in image generation*. The **core insight** of our UiG is to **integrate generative guidance by the strong understanding capabilities during the reasoning process, thereby mitigating the limitations of generative abilities**. To achieve this, we introduce "*Image Editing*" as a bridge to infuse understanding into the generation process. Initially, we verify the generated image and incorporate the understanding of unified models into the editing instructions. Subsequently, we enhance the generated image step by step, gradually infusing the understanding into the generation process. Our UiG framework demonstrates a significant performance improvement in text-to-image generation over existing text-to-image reasoning methods, *e.g.*, a **3.92%** gain on the long prompt setting of the TIIF benchmark. *The project code is available in the Supplementary Materials.*

## 1 Introduction

Recent text-to-image reasoning methods have achieved notable progress in text-to-image generation. These methods generally employ CoT to enhance the image generation process. There are two primary categories of these reasoning: **(1) verification-based reasoning**, which verifies and selects the generated images using CoT, and **(2) prompt-based reasoning**, which enhances the input prompts through CoT. However, both of them separate the understanding and generation, leading to ineffective guidance for image generation from their understanding capabilities.

First, for verification-based reasoning (*e.g.*, ImageCoT (Zhang et al., 2025a)), as illustrated in Figure 1 **(a)**, this method constructs multiple generative branches through repeated sampling and assesses the generative potential throughout the generation process. Specifically, the state of each branch is evaluated to determine whether it still has the potential to continue generating. If the assessment gives a negative result, the branch is terminated. Here, the understanding of the unified model is applied exclusively to the verification of intermediate images, selecting the best output from among numerous samples. However, in this framework, the understanding capability is employed merely as a tool for validation and filtering, rather than providing effective guidance during the generation. As a result, **the generative ability remains confined to scopes that can be reached through repeated sampling**. As shown in Figure 1 **(a)**, given the input prompt, `["There is a cup positioned behind the woman"]`, all outputs consistently place the cup in front of the woman. Consequently, even the best result obtained through verification-based reasoning fails to satisfy the spatial relationship specified in the input prompt.

For the prompt reasoning method (*e.g.*, T2I-R1 (Jiang et al., 2025a)), as shown in Figure 1 **(b)**, CoT is employed to analyze the original prompt from multiple perspectives, *e.g.*, the subject, scene requirements, and other relevant aspects, in order to derive a more refined prompt. Prompt reasoning

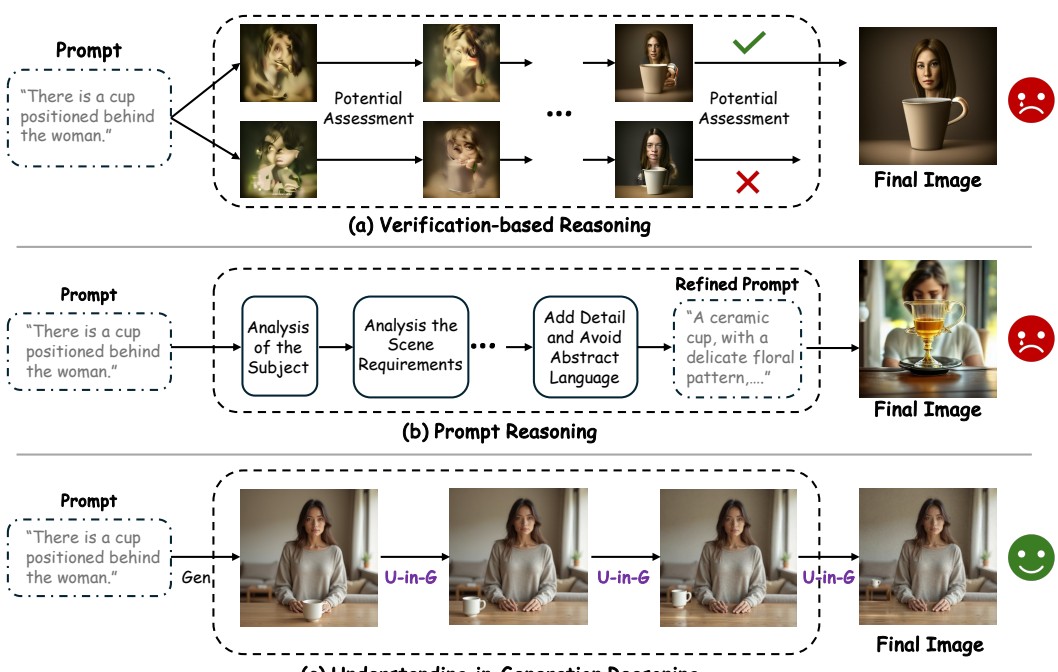

Figure 1: The overall of **(a)** verification-based reasoning, **(b)** prompt reasoning, and **(c)** our UiG.

fully leverages the understanding capabilities of the unified model to enhance the initial prompt, and the resulting refined prompt is then used as the final input for image generation. This approach emphasizes the "*understand first, generate later*" pipeline, in which the understanding phase precedes the generation process. **However, this pipeline focuses exclusively on language during the reasoning and lacks interaction with the generated images, which leads to a failure to capture the inherent limitations of the generation model**. As illustrated in Figure 1 **(b)**, although a refined prompt is obtained through reasoning, the process does not engage with generation, thereby preventing the recognition of generative weaknesses, *e.g.*, the spatial relationship between the cup and the woman. As a result, the refined prompt fails to guide the reasoning to the correct direction.

In this paper, we introduce **Understanding-in-Generation**, an effective text-to-image reasoning framework for unified models that **leverages their strong understanding capabilities to guide and enhance image generation**. The '*out-of-the-box*' insight of our proposed UiG is to integrate generative guidance through the strong understanding capabilities during the reasoning process, thereby addressing the limitations of generative abilities. Meanwhile, our UiG demonstrates the promising potential of unified models to **enhance generation through understanding, powered by using the "*Image Editing*" bridge to infuse understanding into generation**.

Specifically, as illustrated in Figure 1 **(c)**, we first generate an initial image from the original prompt, which shows the generative ability of the unified model. We then evaluate the generated image and recognize its weaknesses, utilizing the understanding capabilities of the unified model. This result is subsequently integrated into the editing prompt, which carries the robust understanding capabilities of the unified model. Finally, we utilize the edited prompt to infuse this understanding into the generation process, guiding the generative direction and overcoming the limitations of the initial image. In the reasoning process, we treat "*Image Editing*" as the bridge to **incorporate understanding into generation, effectively steering the reasoning process in the correct direction and significantly enhancing the generative capability for the unified model**.

Our UiG demonstrates *significant improvement* for the generative capabilities of unified models, leveraging their robust understanding capabilities. Compared with the previous text-to-image reasoning methods, our UiG achieves significant performance gains on both the TIIF and WISE benchmarks. Our key contributions are summarized as follows:

- We propose Understanding-in-Generation, an effective reasoning framework to mitigate the limitation of generative capabilities via infusing the understanding guidance.

- We demonstrate the promising potential of unified models to reinforce their generative capabilities through their strong understanding capabilities during reasoning.
- Our UiG demonstrates a substantial performance gain over existing text-to-image reasoning methods, *e.g.*, a 3.92% gain on the long prompt setting of the TIIF benchmark.

## 2 RELATED WORK

### 2.1 UNDERSTANDING-GENERATION UNIFIED MODEL

Recent multimodal large language models (Driess et al., 2023; Peng et al., 2023; Alayrac et al., 2022; Zhu et al., 2023b; Wang et al., 2024a; Bai et al., 2025; Chen et al., 2024; Li et al., 2023; Lin et al., 2023; Gao et al., 2023; Zhang et al., 2023) have demonstrated remarkable progress in visual understanding powered by the strong vision encoders (Dosovitskiy et al., 2020; Radford et al., 2021; Girdhar et al., 2023; Zhu et al., 2023a; Guo et al., 2023; Lyu et al., 2024). Building on this success, unifying visual understanding and generation within a single framework has emerged as a central research frontier. Early unified models, such as Transfusion (Zhou et al., 2024), Chameleon (Team, 2024), Emu3 (Wang et al., 2024b), and Show-o (Xie et al., 2024), employ visual tokenizers (*e.g.*, VQ-VAE (Van Den Oord et al., 2017)) to encode images into sequences of tokens, to enable seamless multimodal understanding and generation further. However, these discrete visual tokens introduce visual information loss, limiting the extraction of fine-grained semantic content. To address this, the Janus (Chen et al., 2025b) series decouples visual encoding for understanding and generation by adopting separate encoders, though task conflicts within the shared LLM parameter space can hinder its performance. Meanwhile, BAGEL (Deng et al., 2025) adopts a Mixture-of-Experts architecture, assigning autoregressive text generation and diffusion-based image synthesis to distinct components. Despite their effectiveness, these models still struggle to fully exploit their strong understanding capabilities during the generative process, which restricts their ability to produce images with complex logical content.

### 2.2 CHAIN-OF-THOUGHT FOR TEXT-TO-IMAGE GENERATION

Chain-of-Thought (Wei et al., 2022) has played an effective role in LLM (Xia et al., 2024; Zhang et al., 2024; Deng et al., 2024; Chen et al., 2025a; Kang et al., 2025; Wu et al., 2025b; Wang & Zhou, 2024) and MLLM (Wang et al., 2025; Xu et al., 2024; Shao et al., 2024; Zhao et al., 2025; Ma et al., 2025; Jiang et al., 2025b), enabling them to decompose complex tasks into structured intermediate steps. Recent research has extended CoT into text-to-image generation, primarily through verification-based and prompt-based approaches. Verification-based methods (*e.g.*, Image-CoT (Zhang et al., 2025a)) generate multiple candidate images via repeated sampling and then apply CoT to verify intermediate results. However, in this setting, reasoning functions solely as a verification stage, leaving the generative process unguided. Consequently, the generative capacity remains restricted to outcomes accessible through repeated sampling. In contrast, prompt-based methods (*e.g.*, T2I-R1 (Jiang et al., 2025a), ReasonGen-R1 (Zhang et al., 2025b), and ImageGenCoT (Liao et al., 2025)) leverage CoT to refine the input prompt by decomposing it into semantic aspects, to further improve the prompt for generation. However, this reasoning operates independently of the generation process and cannot overcome the limitations of generative capability.

## 3 METHODOLOGY

### 3.1 PROBLEM FORMULATION

Text-to-image generation aims to synthesize the image $I$ from the given text prompt $t$ by the generative model $f$:

$$I = f(t). \tag{1}$$

However, the generative models struggle with complex prompts requiring compositional understanding and spatial reasoning. To address this, the reasoning methods are proposed for text-to-image generation to decompose the generation process into sequential reasoning steps:

$$h_i = \begin{cases} Reasoner\,(\,t, \phi\,), & i \in \{1\} \\ Reasoner\,(\,t, h_{i-1}\,), & i \in \{2, 3, ..., n\}, \end{cases} \tag{2}$$

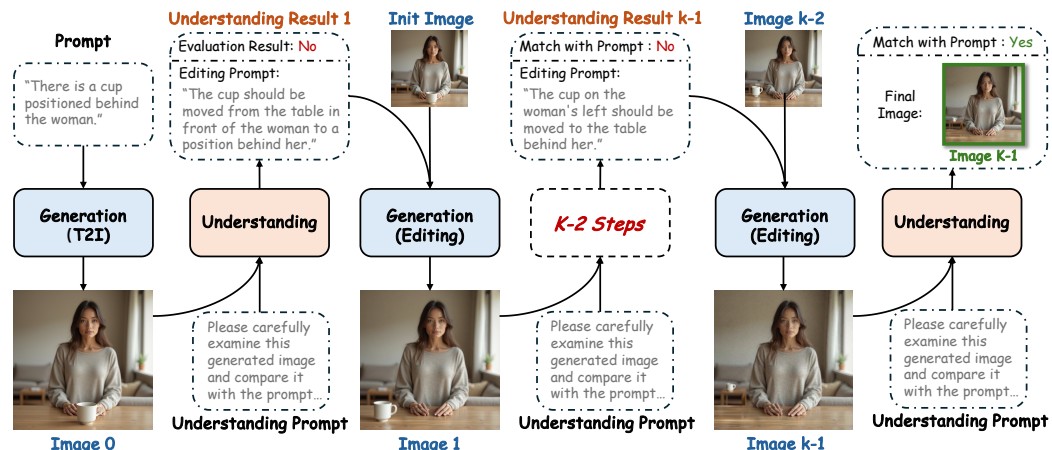

Figure 2: The overall framework of our proposed Understanding-in-Generation reasoning.

where $Reasoner(\cdot)$ is the reasoning pipeline for each step, and $h_i$ is the intermediate stage of step $i$. Finally, the final image $I$ is then conditioned on the complete reasoning chain:

$$I = f(t \mid \{h_1, h_2, ..., h_n\}).\qquad(3)$$

### 3.2 THE UNDERSTANDING-IN-GENERATION REASONING

We present Understanding-in-Generation reasoning in Figure 2. UiG enhances the generative capabilities of unified models by infusing their strong understanding abilities into the step-by-step reasoning process. The core insight of UiG is to leverage the powerful understanding capabilities of unified models to provide effective guidance throughout the generative reasoning steps. Specifically, given an input text prompt $Prompt$, we first generate an initial image $Image_0$ using the original text-to-image generative function $\text{Generate}_{t2i}(\cdot)$ of the unified model.

Next, we combine the initial image $Image_0$ with an understanding prompt $Prompt^{\text{un}}$ (*see Appendix for the full prompt*) as input to the understanding module. This gives an evaluation result $R_1^{\text{match}}$ and an editing prompt $Prompt_1^{\text{edit}}$, formulated as:

$$R_i^{\text{match}},\ Prompt_i^{\text{edit}} = \text{Understand}\left(\text{Combine}\left\{Image_{i-1},\ Prompt^{\text{un}}\right\}\right),\qquad(4)$$

where $\text{Understand}(\cdot)$ denotes the understanding function of the unified model, and $i$ is the reasoning step index (with $i = 1$ in this case).

At the end of each reasoning step, we first examine the evaluation result $R_i^{\text{match}}$. If the response from the understanding module is "Yes", which indicates that the generated image is well-aligned with the given prompt, then the reasoning process is considered complete. Conversely, a "No" response means that further refinement is needed, and the model continues to leverage understanding to enhance the generative output.

As illustrated in Figure 2, we incorporate the strong understanding capability into the generation process via the following formulation:

$$Image_i = \text{Generate}_{editing}\left(\text{Combine}\left\{Image_{i-1},\ Prompt_i^{\text{edit}}\right\}\right),\qquad(5)$$

where $\text{Editing}(\cdot)$ denotes the *image editing* function of the unified model. We use "*Image Editing*" as the bridge to incorporate understanding into generation, effectively steering the reasoning process in the correct direction and significantly enhancing the generative capability for the unified model.

The reasoning process iteratively follows the above steps, ultimately producing the final output image $Image_{\text{final}}$, determined by:

$$Image_{\text{final}} = Image_{k-1},\quad \text{if}\ \left(R_k^{\text{match}} = \text{Yes}\right),\qquad(6)$$

Our proposed UiG significantly improves the generative capabilities of unified models by effectively leveraging their robust understanding strengths to guide the generation. Lastly, we evaluate our UiG on both the TIIF and WISE benchmarks, and the experiment results demonstrate substantial performance gain over existing text-to-image reasoning methods.

### 3.3 WHY WE NEED UNDERSTANDING IN GENERATION?

We explore the intuition behind our proposed UiG by addressing a central question: *"Why is understanding helpful in generation?"* As demonstrated in Figure 3 **(a)**, the Understanding-Generation-Separation reasoning approach utilizes a Chain-of-Thought strategy to decompose the prompt construction process into discrete components, *e.g.*, subject, scene requirements, and details. This approach follows a *"first understand, then generate"* paradigm, in which understanding and generation are explicitly decoupled. However, this separation restricts the reasoning process from incorporating the generative limitations of the base model. Consequently, as depicted in Figure 3 **(a)**, this reasoning fails to guide prompt refinement in ways that address the generative weakness of the base model, *e.g.*, the incorrect

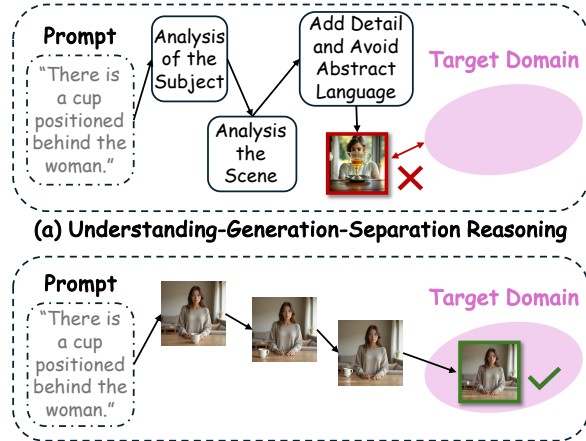

(a) Understanding-Generation-Separation Reasoning

(b) Understanding-in-Generation Reasoning

Figure 3: The comparison of reasoning processes between the **(a)** Understanding-Generation-Separation and **(b)** Understanding-in-Generation reasoning.

spatial relationship between the cup and the woman. In contrast, our Understanding-in-Generation reasoning paradigm leverages the strong understanding capability of the base model to directly guide the reasoning process during generation. As illustrated in Figure 3 **(b)**, the understanding of the model identifies problems in the initial output, *e.g.*, the incorrect relative spatial position between the cup and the woman, and guides the generative direction to overcome these problems. This understanding-in-generation process leads to an output that aligns with the desired target domain.

### 3.4 IMPLEMENTATION

**Base Model:** We utilize BAGEL (Deng et al., 2025) as the base unified model to demonstrate the effectiveness of our proposed UiG framework. BAGEL is a robust and versatile unified model designed with a Mixture-of-Experts architecture. It dedicates specialized components to autoregressive text generation and diffusion-based image generation. Notably, BAGEL has demonstrated superior performance among open-source unified models. To illustrate the significant improvement enabled by our UiG framework, we apply our method to this state-of-the-art open-source model.

**Understanding Prompt:** We have designed an understanding prompt to enhance the cognitive and analytical capabilities of the unified model. In this way, the base model identifies the weaknesses in the generated images and specifies guidance for improvement in the visual editing. The full text of the understanding prompt is provided *in Section A of the Appendix*.

**Iteration:** We set the maximum number of iterations and performed an ablation study on this hyperparameter in Table 3. Our results show that the optimal value for maximum iterations is **4**.

## 4 EXPERIMENT

### 4.1 BENCHMARK AND EXPERIMENT SETUP

**Benchmark:** We evaluate the performance of text-to-image reasoning using the TIIF benchmark (Wei et al., 2025) and the WISE benchmark (Niu et al., 2025). The TIIF benchmark provides a comprehensive framework for fine-grained assessment of text-to-image models. It features 36 novel prompt combinations spanning six compositional dimensions, along with 100 real-world,

Table 1: **Evaluation Results on the TIIF Benchmark** of baselines and our UiG. "Attr." refers to Attribute, "Rela." to Relation, and "Reas." to Reasoning. To facilitate a direct comparison of performance, we set the performance of the SoTA model as -0.00. Gains relative to the SoTA model are indicated in purple, while reductions are highlighted using blue.

| Model | Overall | | Basic Following | | | | Advanced Following | | | | | | Designer |
|---|---|---|---|---|---|---|---|---|---|---|---|---|---|
| | | | Avg. | Attr. | Rela. | Reas. | Avg. | Attr. +Rela. | Attr. +Reas. | Rela. +Reas. | Style | Text | Real World |
| *Short Prompt Setting* | | | | | | | | | | | | | |
| Llamagen | 41.67 | ↓26.92 | 53.00 | 48.33 | 59.57 | 51.07 | 35.89 | 38.82 | 40.84 | 49.59 | 46.67 | 0.00 | 39.73 |
| LightGen | 53.22 | ↓15.37 | 66.58 | 55.83 | 74.82 | 69.07 | 46.74 | 62.44 | 61.71 | 50.34 | 53.33 | 0.00 | 50.92 |
| Show-o | 59.72 | ↓8.87 | 73.08 | 74.83 | 78.82 | 65.57 | 53.67 | 60.95 | 68.59 | 66.46 | 63.33 | 3.83 | 55.02 |
| Infinity | 62.07 | ↓6.52 | 73.08 | 74.33 | 72.82 | 72.07 | 56.64 | 60.44 | **74.22** | 60.22 | 80.00 | 10.83 | 54.28 |
| Janus-Pro | 66.50 | ↓2.09 | 79.33 | 79.33 | 78.32 | **80.32** | 59.71 | 66.07 | 70.46 | 67.22 | 60.00 | 28.83 | 65.84 |
| Bagel | 68.25 | ↓0.34 | 74.84 | 76.50 | 79.00 | 69.00 | 64.58 | 65.83 | 61.88 | 66.41 | **90.00** | **36.20** | 69.40 |
| ImageCoT | 47.41 | ↓21.18 | 61.60 | 63.00 | 63.57 | 58.23 | 48.94 | 60.95 | 46.53 | 46.36 | 40.00 | 1.81 | 46.27 |
| ReasonGen | 61.66 | ↓6.93 | 76.55 | 79.50 | 74.49 | 75.65 | 62.36 | 66.43 | 72.28 | 57.27 | 40.00 | 13.57 | 75.75 |
| T2I-R1 | 68.59 | -0.00 | **82.90** | **86.50** | **83.47** | 78.73 | **69.05** | 71.64 | 72.43 | **69.40** | 60.00 | 27.60 | 67.54 |
| **Our UiG** | **69.70** | ↑1.11 | 80.40 | 84.50 | 80.58 | 76.13 | 67.74 | **73.84** | 66.59 | 65.95 | 80.00 | 30.32 | 69.40 |
| *Long Prompt Setting* | | | | | | | | | | | | | |
| Llamagen | 38.22 | ↓28.97 | 50.00 | 42.33 | 60.32 | 47.32 | 32.61 | 31.57 | 47.22 | 46.22 | 33.33 | 0.00 | 35.62 |
| LightGen | 43.41 | ↓23.78 | 47.91 | 47.33 | 45.82 | 50.57 | 41.53 | 40.82 | 50.47 | 45.34 | 53.33 | 6.83 | 50.55 |
| Show-o | 58.86 | ↓8.33 | 75.83 | 79.83 | 78.32 | 69.32 | 50.38 | 56.82 | 68.96 | 56.22 | 66.67 | 2.83 | 50.92 |
| Infinity | 62.32 | ↓4.87 | 75.41 | 76.83 | 77.57 | 71.82 | 54.98 | 55.57 | 64.71 | 59.71 | 73.33 | 23.83 | 56.89 |
| Janus-Pro | 65.02 | ↓2.17 | 78.25 | 82.33 | 73.32 | 79.07 | 58.82 | 56.20 | 70.84 | 59.97 | 70.00 | 33.83 | 60.25 |
| Bagel | 66.54 | ↓0.65 | 77.09 | **83.00** | 74.54 | 72.72 | 64.67 | 67.90 | 64.01 | 65.81 | 66.67 | 33.03 | 70.15 |
| ImageCoT | 45.41 | ↓21.78 | 59.23 | 59.50 | 56.97 | 61.23 | 44.76 | 53.80 | 41.14 | 45.15 | 40.00 | 3.17 | 47.76 |
| ReasonGen | 65.11 | ↓2.08 | 77.14 | 81.00 | 75.72 | 74.69 | 65.83 | 70.81 | 70.99 | 61.96 | 53.33 | 28.05 | 69.40 |
| T2I-R1 | 67.19 | -0.00 | **81.63** | **83.00** | 79.43 | **82.46** | 68.00 | 69.47 | **69.95** | 70.40 | 63.33 | 26.24 | 60.45 |
| **Our UiG** | **71.11** | ↑3.92 | 79.05 | **83.00** | 75.83 | 78.33 | **70.36** | **73.36** | 67.59 | **73.27** | **76.67** | **36.65** | **75.00** |

designer-level prompts that demand sophisticated aesthetic judgment. The WISE benchmark, a widely adopted standard in this domain, poses challenges by embedding world knowledge within prompts, thereby testing a model's capacity for knowledge-based text-to-image reasoning.

**Baselines:** Our baseline models consist of original AR-based models, including Llamagen (Sun et al., 2024), LightGen (Wu et al., 2025a), Show-o (Xie et al.), Infinity (Han et al., 2025), Janus-Pro (Chen et al., 2025b), Orthus (Kou et al., 2024), Vila-u (Wu et al., 2024), and Emu3 (Wang et al., 2024b); as well as text-to-image reasoning models, including ImageCoT (Zhang et al., 2025a), ReasonGen-R1 (Zhang et al., 2025b), and T2I-R1 (Jiang et al., 2025a).

**Experiment Setting:** We follow the official setting of the TIIF benchmark and WISE benchmark to evaluate all the baselines and our proposed Understanding-in-Generation reasoning. For more details about these two benchmarks, *please refer to Section B in the Appendix*.

## 4.2 QUANTITATIVE RESULTS

We present the evaluation results on the TIIF benchmark in Table 1. As shown in Table 1, our proposed Understanding-in-Generation reasoning framework demonstrates strong performance improvement, achieving a **1.11%** gain under the short prompt setting and a **3.92%** gain under the long prompt setting. These substantial gains highlight the effectiveness of UiG in enhancing the generative capabilities of unified models. Furthermore, we report evaluation results on the WISE benchmark in Table 2. The quantitative findings indicate that UiG exhibits competitive performance in text-to-image reasoning, achieving a **0.16** score gain over existing reasoning methods. The results across both TIIF and WISE benchmarks provide evidence to show the effectiveness of our UiG in reinforcing the text-to-image reasoning for unified models.

## 4.3 QUALITATIVE RESULTS

As illustrated in Figure 4, we present a visual comparison between existing text-to-image reasoning methods (*e.g.*, ImageCoT (Zhang et al., 2025a), ReasonGen-R1 (Zhang et al., 2025b), and T2I-R1 (Jiang et al., 2025a)) and our proposed UiG framework. The visual results demonstrate a notable improvement in prompt alignment achieved by UiG in the generated images. For instance, given the prompt ["On a stool are four black bal-

Table 2: **Evaluation Results on the WISE benchmark**. To facilitate a direct comparison of performance, we set the performance of the SoTA model as -0.00. Gains relative to the SoTA model are indicated in purple, while reductions are highlighted using blue.

| Model | Overall | | Cultural | Time | Space | Biology | Physics | Chemistry |
|---|---|---|---|---|---|---|---|---|
| Janus-1.3B | 0.23 | ↓0.31 | 0.16 | 0.26 | 0.35 | 0.28 | 0.30 | 0.14 |
| Janus-Pro-1B | 0.26 | ↓0.28 | 0.20 | 0.28 | 0.45 | 0.24 | 0.32 | 0.16 |
| Orthus-7B-instruct | 0.27 | ↓0.27 | 0.23 | 0.31 | 0.38 | 0.28 | 0.31 | 0.20 |
| vila-u-7B | 0.31 | ↓0.23 | 0.26 | 0.33 | 0.37 | 0.35 | 0.39 | 0.23 |
| Show-o | 0.35 | ↓0.19 | 0.28 | 0.40 | 0.48 | 0.30 | 0.46 | 0.30 |
| Janus-Pro-7B | 0.35 | ↓0.19 | 0.30 | 0.37 | 0.49 | 0.36 | 0.42 | 0.26 |
| Emu3 | 0.39 | ↓0.15 | 0.34 | 0.45 | 0.48 | 0.41 | 0.45 | 0.27 |
| T2I-R1 | 0.54 | -0.00 | 0.56 | 0.55 | 0.63 | 0.54 | 0.55 | 0.30 |
| Our UiG | **0.70** | ↑**0.16** | **0.74** | **0.62** | **0.74** | **0.62** | **0.76** | **0.61** |

Figure 4: The visual comparison between the existing text-to-image reasoning methods and our proposed Understanding-in-Generation reasoning.

loons and three yellow bananas"], only UiG accurately generates the correct quantities for both balloons and bananas, as shown in Figure 4. Furthermore, in response to the prompt ["A wooden laptop is sitting silently on a wooden desk, not displaying a beautiful scenery"], all existing reasoning methods fail to generate an image without displaying scenery. In contrast, UiG produces a visually accurate result

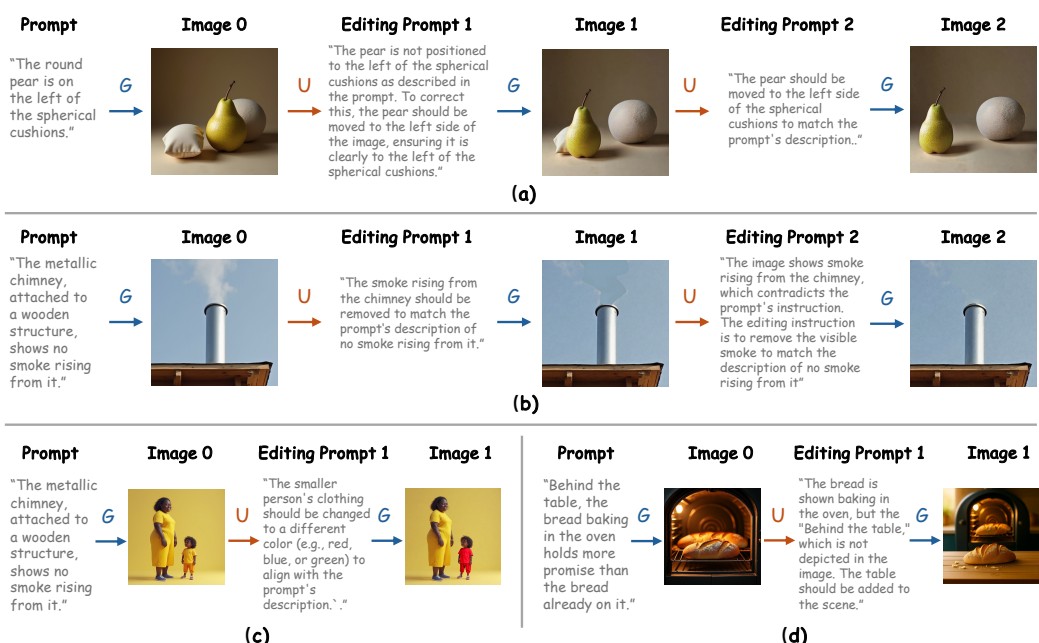

Figure 5: The visualization of the reasoning process within our Understanding-in-Generation framework. **(a)** and **(b)** correspond to the cases with three iterations, while **(c)** and **(d)** represent the two-iteration cases. *Additional visualizations are provided in Section C.2 of the Appendix.*

Table 3: **Ablation on the maximum iteration hyperparameter.** We present an ablation study on the maximum iteration hyperparameter using the TIIF benchmark, varying its value from 1 to 5. The results indicate that a setting of *4 iterations* shows the best performance.

| Maximum Iteration | Overall | Basic Following | | | | Advanced Following | | | | | | Designer |
|---|---|---|---|---|---|---|---|---|---|---|---|---|
| | | Avg. | Attr. | Rela. | Reas. | Avg. | Attr. +Rela. | Attr. +Reas. | Rela. +Reas. | Style | Text | Real World |
| | | | | | | **Short Prompt Setting** | | | | | | |
| Value = 1 | 64.31 | 76.43 | 76.50 | 75.54 | 78.25 | 66.97 | 73.14 | 65.82 | 69.44 | 66.67 | 7.24 | 67.16 |
| Value = 2 | 67.18 | 77.73 | 82.50 | 81.68 | 69.00 | 65.42 | 72.15 | 62.51 | 65.88 | 70.00 | 26.70 | **74.25** |
| Value = 3 | 69.06 | 79.15 | 84.00 | 80.85 | 72.60 | **69.85** | **77.40** | 64.96 | **72.60** | 73.33 | 23.08 | 72.76 |
| Value = 4 | **69.70** | 80.40 | **84.50** | 80.58 | 76.13 | 67.74 | 73.84 | 66.59 | 65.95 | **80.00** | **30.32** | 69.40 |
| Value = 5 | 67.96 | **82.62** | 83.00 | **82.53** | **82.33** | 68.12 | 70.30 | **68.51** | 71.78 | 63.33 | 23.08 | 66.79 |
| | | | | | | **Long Prompt Setting** | | | | | | |
| Value = 1 | 65.74 | 75.74 | 78.00 | 79.25 | 69.96 | 65.57 | 70.39 | 65.62 | 65.11 | 73.33 | 15.84 | 67.54 |
| Value = 2 | 68.35 | 78.72 | 78.50 | **82.02** | 75.65 | 67.66 | 72.15 | 62.93 | 70.34 | 73.33 | 25.34 | 72.76 |
| Value = 3 | 68.76 | 79.86 | 86.50 | 77.35 | 75.73 | 68.21 | 70.07 | **70.01** | 68.88 | 70.00 | 29.41 | 70.90 |
| Value = 4 | **71.11** | 79.05 | 83.00 | 75.83 | **78.33** | **70.36** | 73.36 | 67.59 | 73.27 | 76.67 | **36.65** | 75.00 |
| Value = 5 | 67.03 | 77.34 | 81.00 | 76.28 | 74.73 | 66.80 | 69.13 | 66.28 | 70.39 | 63.33 | 27.15 | 75.00 |

consistent with the prompt, as also shown in Figure 4. These visual comparisons show the substantial gains in prompt following and generation accuracy provided by our UiG framework. Additional results are presented *in Section C.1 of the Appendix* to further validate the effectiveness of UiG.

Additionally, we present a visualization of the reasoning process within our UiG framework, as shown in Figure 5. In case **(a)**, UiG effectively identifies a weakness in the original generated image, specifically the incorrect spatial relationship between the pear and the spherical cushions, which is attributed to its strong understanding capability. Subsequently, our UiG uses "*Image Editing*" as a bridge to infuse the understanding into the generation, further to guide the generative direction to refine the recognized weakness in spatial relationship, and finally generate the well-matched image.

## 5 ABLATION STUDY

**Ablation on the Maximum Iteration Parameter.** To determine the optimal value for the maximum iteration hyperparameter, we conducted experiments by varying its value from 1 to 5. The corre-

Table 4: **Ablation on the "*Image Editing*" bridge.** We report the results on the TIIF benchmark with or without using the "*Image Editing*" bridge to infuse the understanding into generation.

| "Image Editing" Bridge | Overall | Basic Following | | | | Advanced Following | | | | | | Designer |
|---|---|---|---|---|---|---|---|---|---|---|---|---|
| | | Avg. | Attr. | Rela. | Reas. | Avg. | Attr. +Rela. | Attr. +Reas. | Rela. +Reas. | Style | Text | Real World |
| | | | | Short Prompt Setting | | | | | | | | |
| ✗ | 64.30 | 77.70 | 81.50 | 80.05 | 71.56 | 65.18 | 69.75 | 65.06 | **66.98** | 70.00 | 10.41 | 63.43 |
| ✓ | **69.70** | **80.40** | **84.50** | **80.58** | **76.13** | **67.74** | **73.84** | **66.59** | 65.95 | **80.00** | **30.32** | **69.40** |
| Δ | ↑5.40 | ↑2.70 | ↑3.00 | ↑0.53 | ↑4.57 | ↑2.56 | ↑4.09 | ↑1.53 | ↓1.03 | ↑10.00 | ↑19.91 | ↑5.97 |
| | | | | Long Prompt Setting | | | | | | | | |
| ✗ | 65.04 | 78.14 | 80.50 | **81.97** | 71.96 | 66.24 | 69.35 | **68.00** | 68.08 | 63.33 | 15.38 | 66.79 |
| ✓ | **71.11** | **79.05** | **83.00** | 75.83 | **78.33** | **70.36** | **73.36** | 67.59 | **73.27** | **76.67** | **36.65** | **75.00** |
| Δ | ↑6.07 | ↑0.91 | ↑2.50 | ↓6.14 | ↑6.37 | ↑4.12 | ↑4.01 | ↓0.41 | ↑5.19 | ↑13.34 | ↑21.27 | ↑8.21 |

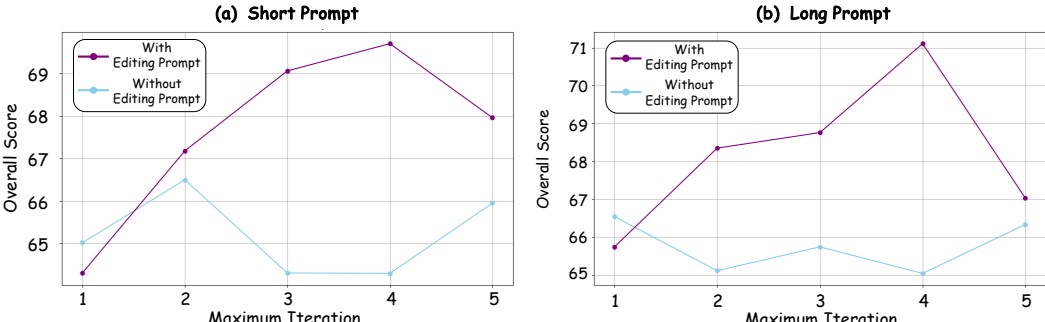

Figure 6: **Ablation Study on the "*Image Editing*" bridge**. We show the trend of the performance with the maximum iteration. **(a)** illustrates the results under the short prompt setting, while **(b)** shows the results under the long prompt setting.

sponding results are presented in Table 3. The results indicate that setting the maximum iteration value to 4 demonstrates the best performance on the TIIF benchmark.

**Ablation on the "*Image Editing*" Bridge.** To assess the effectiveness of the proposed "*Image Editing*" bridge, we design a comparative pipeline that does not infuse understanding into the editing prompt. Specifically, the pipeline uses only the original prompt as guidance to refine the generated image during reasoning, without incorporating any understanding capabilities from the base model. As shown in Table 4, incorporating the editing prompt enriched by the base model's understanding leads to substantial improvements over the baseline pipeline. Furthermore, we examine the performance trends with increasing maximum iterations for both UiG and the pipeline, which lack the "*Image Editing*" bridge. As illustrated in Figure 6, the performance of the baseline pipeline fluctuates with the number of iterations, suggesting that the reasoning process fails to consistently guide image generation. In contrast, the performance curve of UiG shows a steady upward trend, indicating that the "*Image Editing*" bridge successfully integrates understanding into the generation process, and further reinforces the generation in a progressively positive direction.

# 6 CONCLUSION

In this paper, we propose Understanding-in-Generation, an effective reasoning framework for text-to-image generation, which mitigates the limitation of generative capabilities via infusing the understanding guidance. Our UiG treats "*Image Editing*" as the bridge to incorporate understanding into generation, effectively steering the reasoning process in the correct direction and significantly enhancing the generative capability for the unified model. Our UiG demonstrates the promising potential of unified models to reinforce their generation capabilities through their strong understanding capabilities. We evaluate our reasoning method on both the TIIF and WISE benchmarks, and the experiment results show significant performance improvements over the baselines.

**Limitation and Future Work:** Currently, our UiG does not support video generative models. Future work will focus on enhancing the reasoning capabilities for video generation.

REPRODUCIBILITY STATEMENT

The full project code and experimental results on the TIIF and WISE benchmarks are provided in the *Supplementary Materials*. All relevant resources, including the implementation details and experiment data, are made publicly available to ensure the reproducibility of our work.

ETHICS STATEMENT

We show an additional user study *in the Appendix (Section E)*. Our user study was conducted in full compliance with the ICLR Code of Ethics. Prior to the commencement of the study, we obtained formal approval from the institutional review board (IRB) or an equivalent ethics committee at our affiliated university or research institution. All participants provided informed consent before participating, and were made aware of the purpose of the study, the voluntary nature of their involvement, and their right to withdraw at any time without consequence. No personally identifiable information was collected, and all data was anonymized to ensure participant privacy and confidentiality.

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

# Appendix

## Table of Contents

## A  MORE DETAILS OF UNDERSTANDING-IN-GENERATION REASONING

We show the full understanding prompt in Figure 7. To evaluate whether a generated image aligns with its text prompt, the understanding prompt first guides the base model in parsing the semantic structure of the prompt, identifying key elements such as the main subject, setting, visual attributes (*e.g.*, clothing, colors, lighting), and emotional tone. Then the base model should be prompted to map these components directly onto the visual elements present in the image, enabling a structured cross-modal comparison. By explicitly attending to mismatches between the described and rendered attributes, the base model recognizes omissions (*e.g.*, a missing object, a setting, or a background), inconsistencies (*e.g.*, incorrect spatial relationship or incorrect color for some objects), or misrepresentations of mood and style. These observations should then be synthesized into a coherent editing prompt that specifies the required visual adjustments. This process ensures that the editing prompt is grounded in a detailed diagnostic analysis of the weakness of the generated image, thus facilitating well-matched image refinement in subsequent reasoning processes.

## B  MORE DETAILS OF EXPERIMENT

### B.1  MORE DETAILS OF BENCHMARKS

**TIIF Benchmark** is a comprehensive and fine-grained evaluation benchmark specifically designed to assess the instruction-following capabilities of modern Text-to-Image (T2I) models. Unlike prior benchmarks such as COMPBENCH++ (Huang et al., 2025) or GENAI BENCH (Li et al., 2024), which suffer from semantic redundancy, fixed prompt lengths, and coarse evaluation metrics, TIIF-Bench offers a diverse and hierarchically structured set of 5000 prompts that span a broad range of compositional and semantic complexity.

Each prompt in TIIF-Bench is categorized into three difficulty levels—Basic, Advanced, and Designer-Level Following—and is available in both short and long versions to test prompt-length sensitivity. The prompts are systematically constructed through a two-stage pipeline: (i) concept pool extraction across ten dimensions (attributes, relations, and reasoning), and (ii) attribute composition across 36 defined combinations. In addition to traditional evaluation axes (e.g., color, texture, spatial relations, numeracy), TIIF-Bench introduces three novel dimensions: text rendering, style control, and real-world designer prompts, which reflect practical demands and aesthetic nuance.

TIIF-Bench also introduces a fine-grained, attribute-specific evaluation protocol using large vision-language models (VLMs) like GPT-4o and Qwen-VL2.5. This protocol poses structured yes/no queries to assess semantic alignment without relying on full prompt inclusion, thereby mitigating hallucination effects. For text rendering evaluation, TIIF-Bench proposes a novel metric called Global Normalized Edit Distance (GNED), which robustly measures typographic accuracy by penalizing both over-generation and omission.

Extensive benchmarking across T2I models demonstrates that TIIF-Bench provides deeper diagnostic insight into model robustness, semantic comprehension, and generation fidelity, making it a valuable tool for guiding the development and evaluation of next-generation T2I systems.

**WISE Benchmark** is a novel and scalable benchmark framework designed to evaluate the open-ended instruction-following capabilities of unified models. Recognizing the limitations of prior benchmarks—such as constrained scope, limited real-world diversity, and reliance on human annotations—WISE introduces a fully automated pipeline for constructing realistic, diverse, and instruction-rich evaluation datasets at scale.

WISE leverages web-instructed synthetic data generation by crawling diverse and naturally occurring human instructions from the web and pairing them with image-text datasets. These pairings are used to create visually grounded instruction-following examples across a wide array of domains, including science, daily life, medical scenarios, design, and social situations. The benchmark is distinguished by its ability to test higher-order reasoning, commonsense understanding, multi-step inference, and factual grounding in real-world contexts.

A key feature of WISE is its evaluation strategy, which is both automatic and semantically aware. It employs GPT-4-based preference comparisons, where two candidate model responses are assessed in terms of helpfulness, correctness, and relevance to the given instruction. This approach avoids

**Understanding Prompt**

Please carefully examine this generated image and compare it with the original prompt:

"{original_prompt}"

Analyze the following aspects:

    1. Does the image accurately represent the main subject described in the prompt?
    2. Are the visual details (clothing, environment, style, etc.) consistent with the prompt?
    3. Is the overall mood and atmosphere matching the intended description?
    4. Are there any missing elements or incorrect interpretations?

If the image matches the prompt well, respond with: "MATCH: The image successfully represents the prompt."

If there are discrepancies, respond with: "EDIT_NEEDED: [specific editing instructions]"
For example: "EDIT_NEEDED: The character should be wearing a red dress instead of blue, and the background should be a forest not a city."

Please be specific about what needs to be changed:

Figure 7: The full understanding prompt

rigid ground-truth matching and instead focuses on instructional fidelity and user-centric value, closely aligning with real-world deployment conditions.

WISE is constructed at scale, encompassing over 20K samples with corresponding instructions and reference answers, enabling robust and statistically meaningful evaluation across diverse instruction types and complexity levels. Experimental results presented in the paper show that WISE can differentiate model capabilities more effectively than prior benchmarks, revealing nuanced weaknesses in current unified models that are otherwise missed by narrower evaluation methods.

### B.2 MORE DETAILS OF EXPERIMENT SETUP

All reported experiments in this paper were conducted on NVIDIA A100 GPUs. To ensure fair comparisons, we set the random seed to a fixed value of 42. For further code details, please refer to the complete code provided in our *Supplementary Materials*.

## C ADDITIONAL VISUAL RESULT

### C.1 MORE VISUAL COMPARISON

We present more visual comparison in Figure 8 9 10. The additional visual comparisons demonstrate the significant performance improvement in prompt following and generation accuracy provided by our UiG framework, compared with existing text-to-image reasoning methods.

| Prompt | Our UiG | ImageCoT | ReasonGen | T2I-R1 | BAGEL |
|---|---|---|---|---|---|

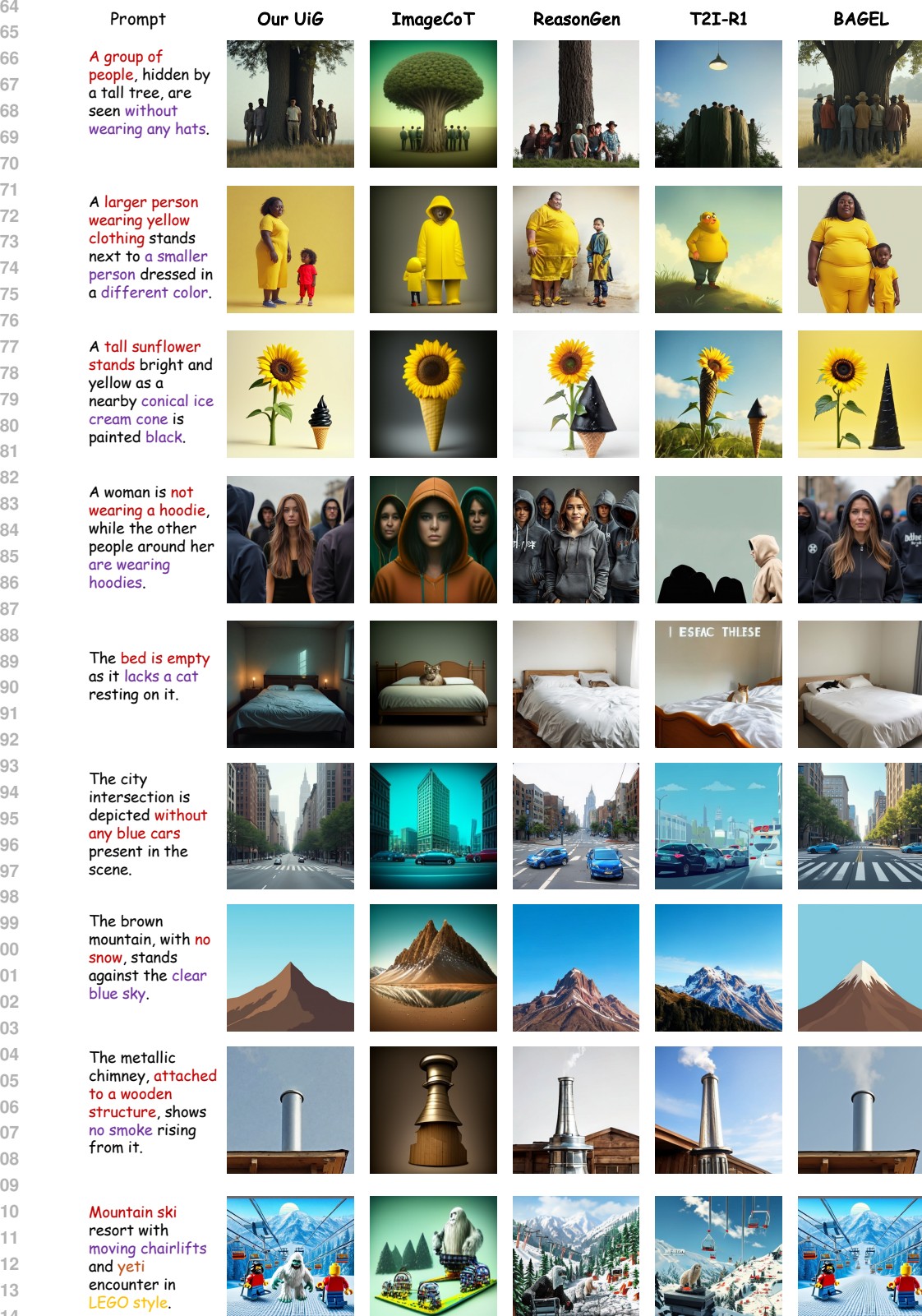

A group of people, hidden by a tall tree, are seen without wearing any hats.

A larger person wearing yellow clothing stands next to a smaller person dressed in a different color.

A tall sunflower stands bright and yellow as a nearby conical ice cream cone is painted black.

A woman is not wearing a hoodie, while the other people around her are wearing hoodies.

The bed is empty as it lacks a cat resting on it.

The city intersection is depicted without any blue cars present in the scene.

The brown mountain, with no snow, stands against the clear blue sky.

The metallic chimney, attached to a wooden structure, shows no smoke rising from it.

Mountain ski resort with moving chairlifts and yeti encounter in LEGO style.

Figure 8: Additional visual comparison with short prompts.

**Prompt**: Nestled in the comforting shadows and silence of the kitchen, the sleek and graceful form of a cat is quietly positioned beneath the looming and functional structure of the microwave, its presence both enigmatic and serene.

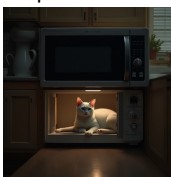 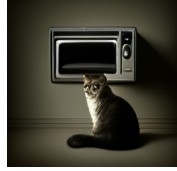 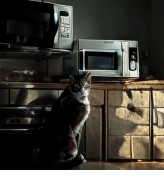 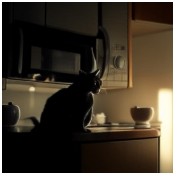 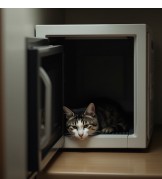

**Our UiG**  **ImageCoT**  **ReasonGen**  **T2I-R1**  **BAGEL**

**Prompt**: In the scene above, one cannot help but notice the overwhelming presence of fish, whose numbers far exceed those of the frogs, which in turn accentuates and brings into sharp relief the vivid contrast between their respective quantities.

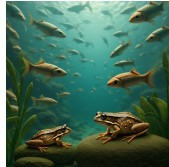 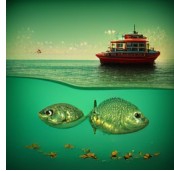 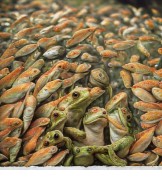 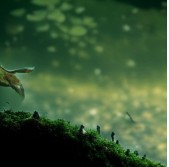 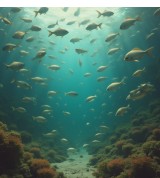

**Our UiG**  **ImageCoT**  **ReasonGen**  **T2I-R1**  **BAGEL**

**Prompt**: Amidst the scene, a larger individual dressed in vibrant yellow clothing stands partially obscured by a smaller figure adorned in a different hue, creating a striking juxtaposition of size and color, as the person in yellow—imbued with the warmth and brightness of a midsummer's day—is intriguingly concealed behind the other, adding an element of mystery and intrigue to the tableau before our eyes.

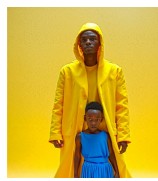 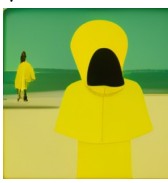 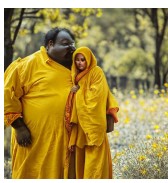 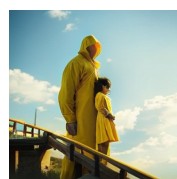 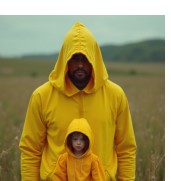

**Our UiG**  **ImageCoT**  **ReasonGen**  **T2I-R1**  **BAGEL**

**Prompt**: The polished, silver fork hovers with poised elegance above the thick, succulent steak, its glinting tines suspended in a moment of hesitation, refraining from the act of piercing through the tender surface, as if respecting the tacit boundary that separates the unmarred from the touched, maintaining its pristine stance while the tangible anticipation lingers above, waiting to complete its intended mission.

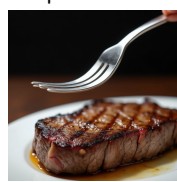 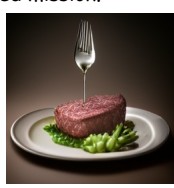 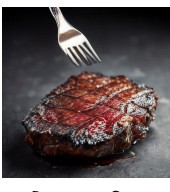 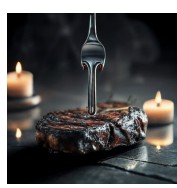 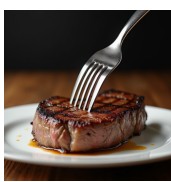

**Our UiG**  **ImageCoT**  **ReasonGen**  **T2I-R1**  **BAGEL**

**Prompt**: In the unfolding tableau, where each element plays its role with silent precision, the windbreaker, with its sleek, weather-resistant fabric, exists conspicuously separate from its woolen counterpart, the sweater, such that it is not draped over, layered upon, nor entwined in any way with the soft garment underneath, instead maintaining its distinct and separate position..

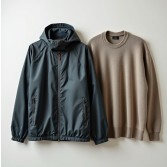 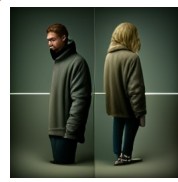 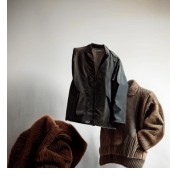 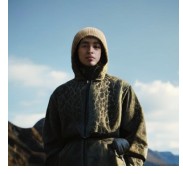 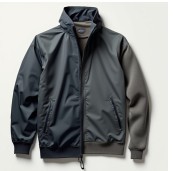

**Our UiG**  **ImageCoT**  **ReasonGen**  **T2I-R1**  **BAGEL**

Figure 9: Additional visual comparison with long prompts (group 1).

**Prompt**: Illuminated by the gentle glow of the afternoon sun streaming through the window, the perfectly spherical soccer ball, with its intricate black and white pentagonal design that seems to invite both casual play and serious competition, sits dutifully in the quiet room, right in front of the imposing yet plush cubic ottoman, whose solid form and soft upholstery stand as a comforting presence in the cozy corner of the living space.

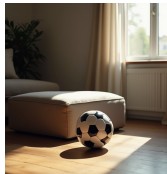 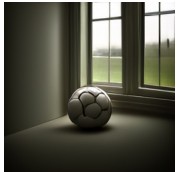 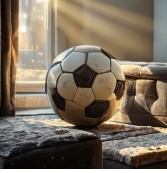 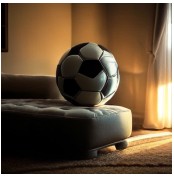 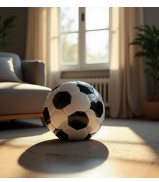

**Our UiG**     **ImageCoT**     **ReasonGen**     **T2I-R1**     **BAGEL**

**Prompt**: The image features a whimsical, hand-drawn illustration centered around a vintage-style teacup with a curved handle and a dainty lid, positioned beside a generously sliced cake adorned with crumpled frosting and layered sponge. Both objects rest on a textured, off-white surface resembling parchment, bathed in soft, golden-hour lighting that casts faint, elongated shadows, enhancing the warmth of the scene. The teacup's ceramic body has a matte finish with subtle speckles, while the cake's maroorued base and crumbly top suggest a traditional recipe. Warm, earthy tones dominate the palette—terracotta, mustard yellow, and sage green—with hints of burnt sienna in the teacup's lac and caramel-brown icing on the cake. Scattered around the central subjects are seven minimalist, cursive words in varying sizes: "drink," "tea," "eat," "cake," "relax," "enjoy," and "delight." Each word is neatly inscribed in a creamy ivory hue, with delicate flourishes at the ends of letters, ensuring readability without overpowering the design. The background blends a pale beige gradient with faint, wispy brushstrokes, evoking a cozy, rustic charm. The overall aesthetic balances whimsy and sophistication, with crisp lines, balanced proportions, and harmonious spacing between elements, creating a serene yet lively invitation to partake in tea and dessert.

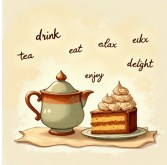 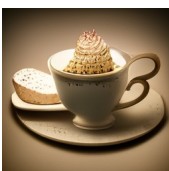 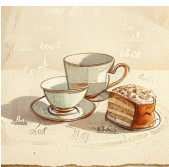 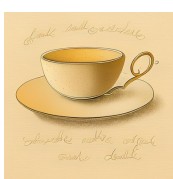 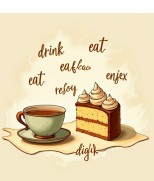

**Our UiG**     **ImageCoT**     **ReasonGen**     **T2I-R1**     **BAGEL**

**Prompt**: The exquisitely crafted oval mirror, with its ornate frame glistening in the ambient light, stands elegantly positioned behind the resplendent diamond engagement ring, whose dazzling facets catch the eye and reflect a myriad of sparkling hues in the mirror's polished surface.

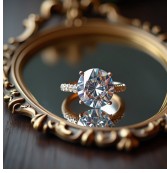 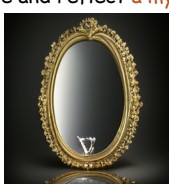 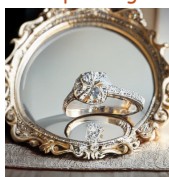 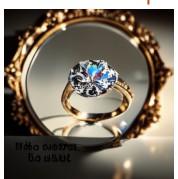 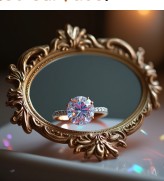

**Our UiG**     **ImageCoT**     **ReasonGen**     **T2I-R1**     **BAGEL**

**Prompt**: At the very bottom of the stage, a microphone—surrounded by a charmingly peculiar assembly of three loaves of bread and two camels—managed to capture the audience's attention and entertain them, weaving a whimsical tapestry of intrigue and amusement as they all worked together in delightful harmony.

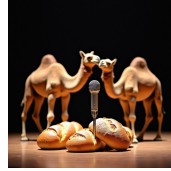 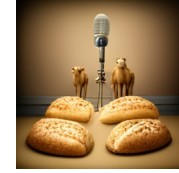 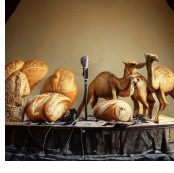 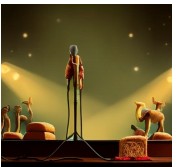 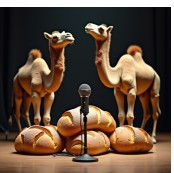

**Our UiG**     **ImageCoT**     **ReasonGen**     **T2I-R1**     **BAGEL**

Figure 10: Additional visual comparison with long prompts (group 2).

## C.2 MORE VISUALIZATION OF REASONING PROCESS

We show additional visualization of the UiG reasoning process in Figure 11 12 13 14. The visualization illustrate the full reasoning process of our proposed UiG, which demonstrates the effectiveness of our UiG to infuse understanding capabilities into the generation step by step.

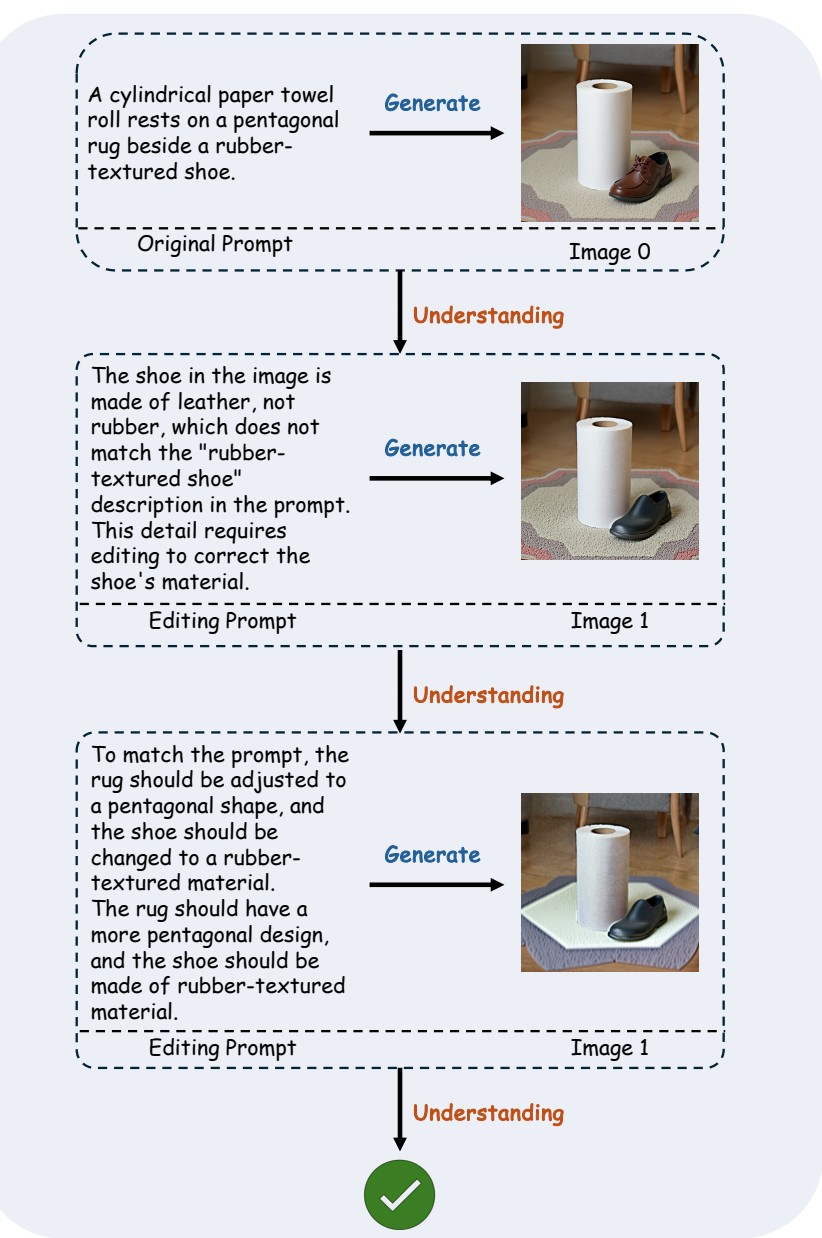

Figure 11: Additional visualization of the reasoning process (group 1).

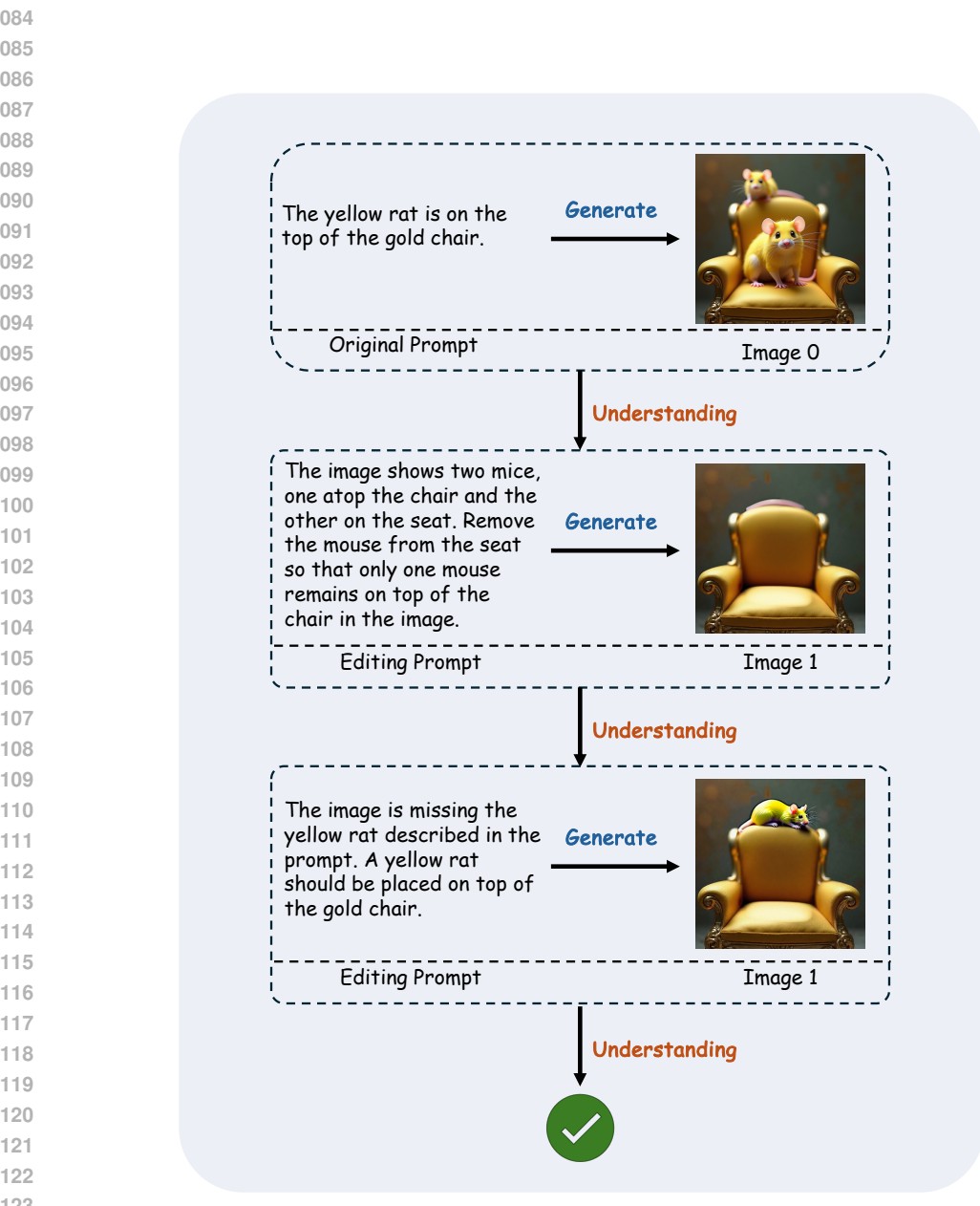

Figure 12: Additional visualization of the reasoning process (group 2).

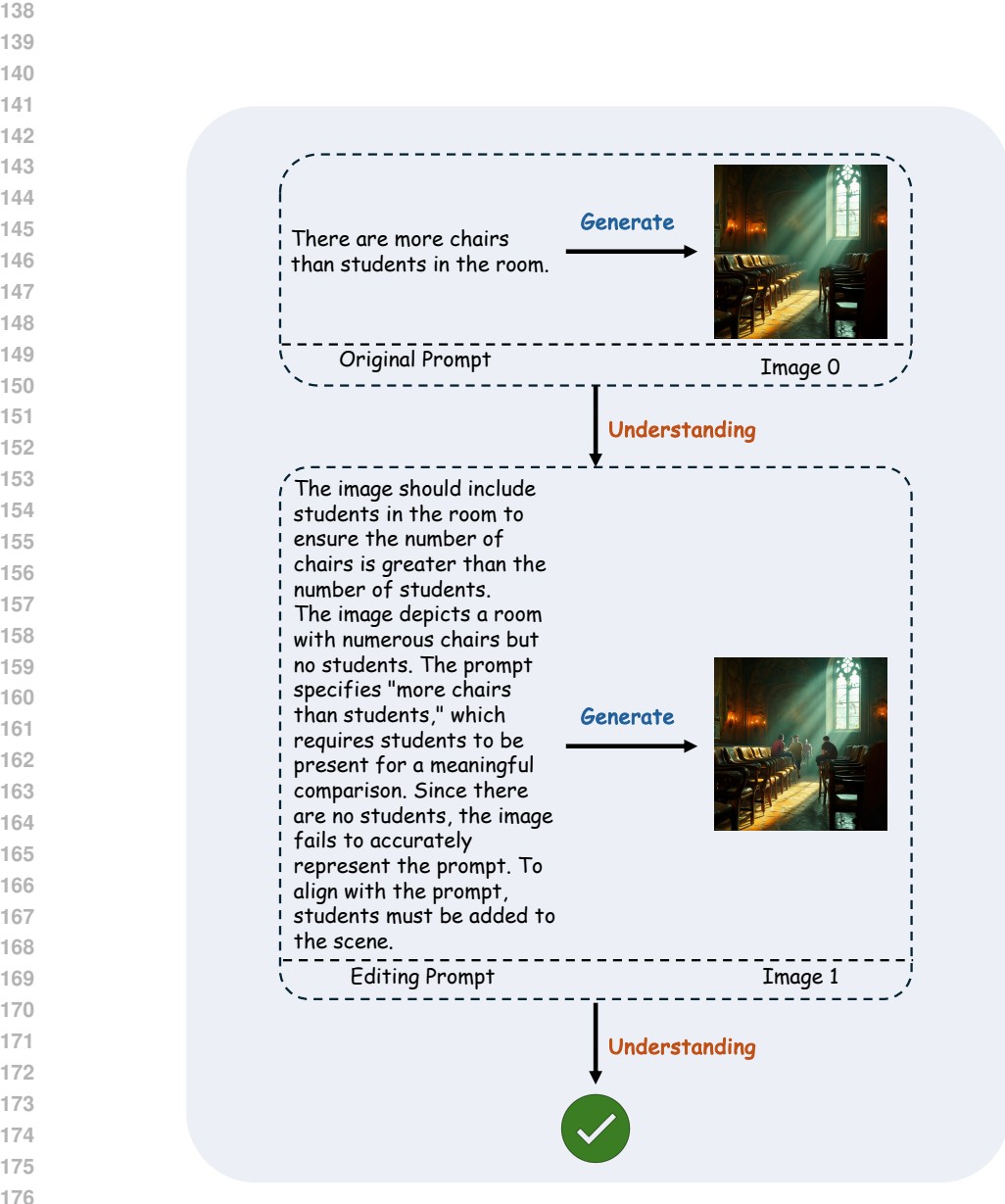

Figure 13: Additional visualization of the reasoning process (group 3).

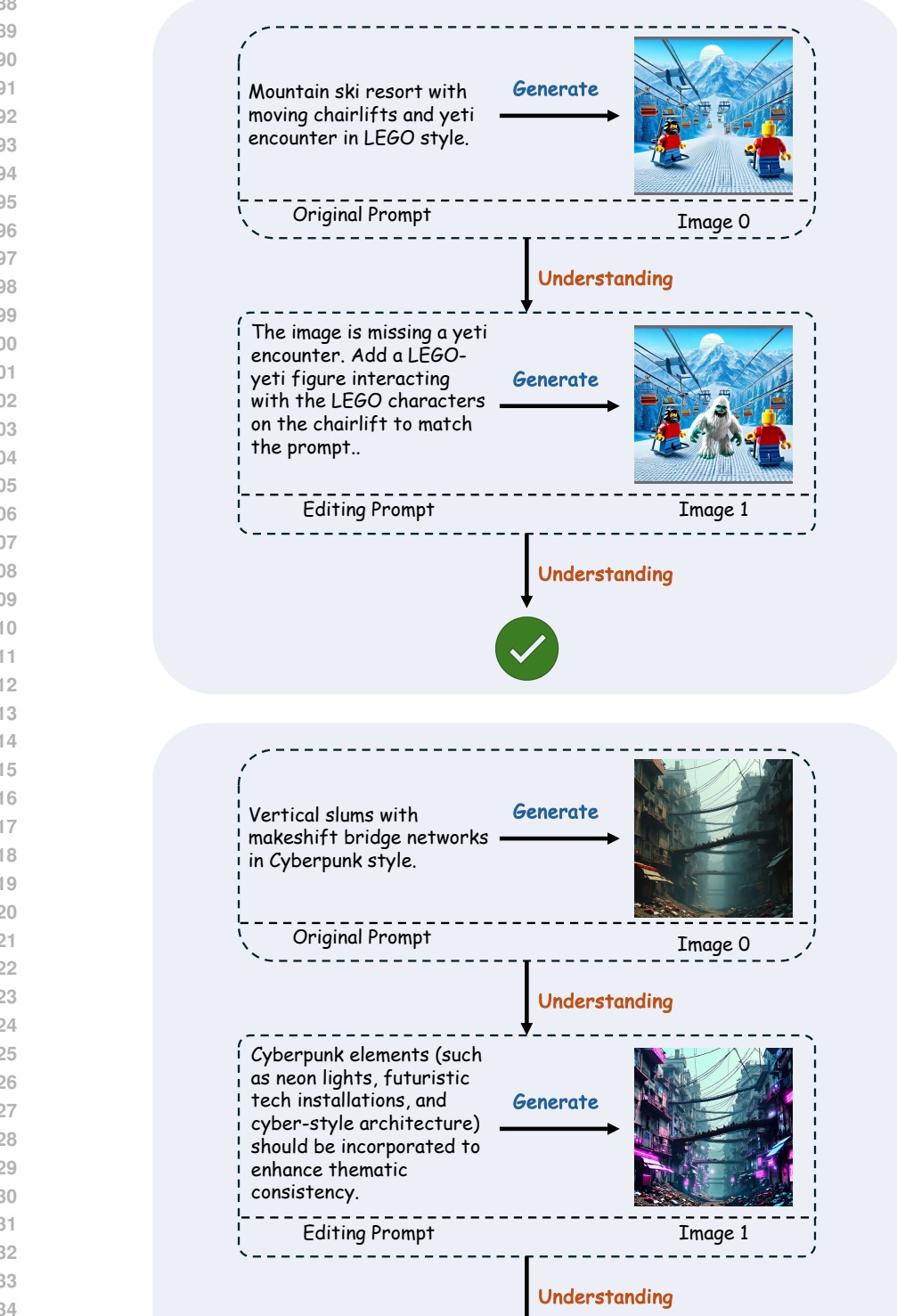

Figure 14: Additional visualization of the reasoning process (group 4).

**Prompt**: An airplane positioned in front of a giraffe.     **Prompt**: A girl stands with a key behind her.

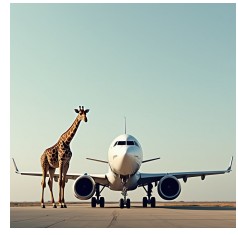 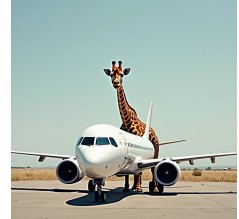 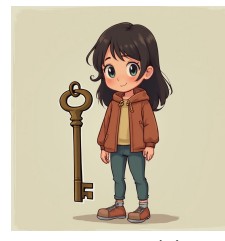 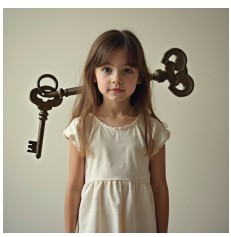

Base Model           Our UiG           Base Model           Our UiG

(a)                                    (b)

Figure 15: The failure cases.

Table 5: **Analysis of the Trade-off**. We analyze the trade-off between performance and latency on the long prompt setting of the TIIF benchmark.

| Model | Performance ↑ | Average Latency (s) ↓ |
|---|---|---|
| Verify-based Reasoning | 67.18 | 391.92 |
| Prompt Reasoning | 67.90 | **108.23** |
| Our UiG | **71.11** | 157.61 |

# D  ADDITIONAL DISCUSSION

## D.1  FAILURE CASE

Our UiG demonstrates a *significant improvement* in the generative capabilities of unified models, leveraging their robust understanding capabilities. However, due to the limitations in the spatial understanding capabilities of the unified model, generating images that depict rare spatial relationships between objects can lead to certain "inconsistencies". As shown in Figure 15 **(a)**, while the UiG successfully generates the correct positional relationship between the giraffe and the airplane (*the giraffe is behind the airplane*) compared to the base model, the relative size relationship between the giraffe and the airplane appears inconsistent. Similarly, as illustrated in Figure 15 **(b)**, compared to the base model, the UiG still generates the image that aligns with the text prompt. However, the keys behind the little girl remain in a floating" state, defying physical laws. This limitation will be addressed as the spatial understanding capabilities of the base model improve. Future work will focus on enhancing the coordination of complex spatial relationships.

## D.2  TRADE-OFF

As shown in Table 5, we analyze the trade-off between performance and latency across verification-based reasoning, prompt reasoning, and our proposed UiG. To ensure a fair comparison, the analysis is conducted using the same base model (BAGEL (Deng et al., 2025)). Our UiG achieves the best performance with lower latency in inference time compared to verification-based reasoning, although it lags slightly behind prompt reasoning in terms of performance. These results demonstrate the overall advantage of UiG in balancing performance and latency.

# E  USER STUDY

To further evaluate the text-to-image reasoning capabilities for baselines and our UiG, we conducted a user study focused on participants' subjective assessments of generation quality and the alignment between generated images and the given prompts.

**Participants.** The study involved 20 participants, 35% of whom were aged 18–25 and 65% aged 25–35. The gender distribution was 60% male and 40% female. Notably, 85% of the participants reported prior experience with AI technologies.

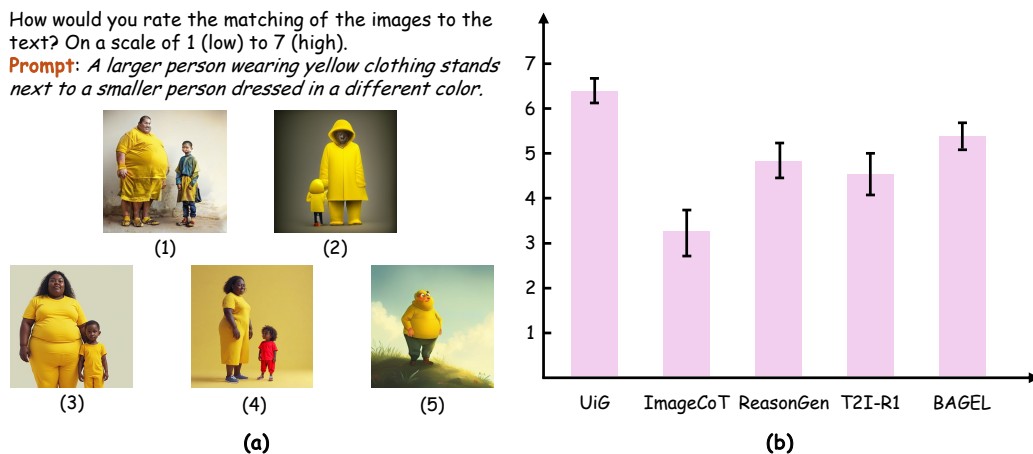

Figure 16: The user study. **(a)** An example of the question. **(b)** The results of the user study.

**Task and Measurement.** As illustrated in Figure 16 **(a)**, participants were asked to rate a series of images on a 7-point Likert scale, evaluating both visual quality and the depth of reasoning relative to the associated input prompts. Images generated by various models were presented in randomized order to control for order effects.

**Results.** As shown in Figure 16 **(b)**, participants consistently rated the quality and reasoning accuracy of images generated by our UiG framework, with the highest mean scores across all comparisons. This indicates a significant performance advantage over baseline models. Furthermore, UiG demonstrated relatively low variance in ratings, underscoring its consistency and reliability in text-to-image reasoning. Taken together, the combination of high average scores and reduced variability suggests that our proposed UiG not only produces more accurate and contextually appropriate images but also does so more consistently across diverse prompts.

## F  THE USE OF LARGE LANGUAGE MODELS (LLM)

We used OpenAI's GPT-4o to assist with the refinement and proofreading of certain sentences in this paper. The LLM was used exclusively to enhance the clarity and coherence of our writing. All content contributions are made by the authors.

