# OpenReview forum: "Understanding-in-Generation: Reinforcing Generative Capability of Unified Model via Infusing Understanding into Generation"
_ICLR.cc/2026/Conference — Submitted to ICLR 2026_

### Official Review · Reviewer_9T57 · 2025-10-29

**Soundness:** 3
**Presentation:** 3
**Contribution:** 2
**Rating:** 4
**Confidence:** 3

**Summary:**

This paper presents a reasoning framework that injecting understanding to the text-to-image generative process. Compared with verification-based and prompt-based generation methods, it no more separates understanding and generation, instead, it treats "image editing" as a bridge to digest understanding into generation by following editing instructions derived from the "understanding prompts". UiG has been extensively evaluated on two benchmarks, achieving 1-3% improvements on TIIF and moderate gains on WISE. Ablations further confirm that iterative reasoning and image editing do contribute to the performance.

**Strengths:**

- Good motivation and problem formulation. Using image editing as a trick to propagate reasoning signals into the generative loop is very interesting and intuitive. It pushes prompt- or sampling-based refinement towards a more promising direction.

- The paper is well presented with good visual examples of progressive reasoning process and generation improvements.

- The method was extensively evaluated on two standard benchmarks with consistent improvements, even though some of them are moderate.

**Weaknesses:**

- The editing iterations intuitively bring computational overhead. The authors should also report "latency" as one of the evaluation metrics.

- The baselines are relatively weak. Comparisons only include open-source models and reasoning variants. It is more insightful to compare with stronger LLMs such as GPT-4o.

- The technical novelty is still limited. The understanding signals are derived from prompt engineering without learning objectives and optimizations, which remain heuristic. It is more encouraging to see that the reasoning chain or signal serves as a real-time critic in the generative process.

**Questions:**

- Does the performance gain mainly from iterative editing or from the understanding capability? What if you use a weaker and smaller LLM to generate understanding instructions?

- Instead of using LLM-based evaluation metric, is there any human-defined metrics for evaluation?

---

### Official Review · Reviewer_qZzK · 2025-10-30

**Soundness:** 3
**Presentation:** 3
**Contribution:** 2
**Rating:** 4
**Confidence:** 4

**Summary:**

The paper proposes Understanding-in-Generation, an effective reasoning framework designed to mitigate the limitations of generative capabilities by infusing understanding guidance.

**Strengths:**

* The writing is clear and well-organized.
* UiG weaves “understanding” into the generation loop—detecting errors on the fly and issuing edit commands that instantly fix spatial slips like “the cup should be behind the woman,” eliminating the old detach-then-filter pipeline.

**Weaknesses:**

* The authors claim to introduce image editing as a bridge connecting generation and understanding. However, this process seems cumbersome. Normally, once a model generates an image from a prompt or through prompt reasoning, the generation process ends. In this paper, however, the model undergoes multiple rounds of image generation → image understanding → image generation, which seems repetitive.
* In Figure 1, the authors discuss logical/understanding problems that arise during generation. This is indeed a common and important issue. I’m curious about:
   1. The logical/understanding problems in the generation process are diverse and complex—some are difficult to describe even in language. Is it really effective to use an MLLM to evaluate and identify these issues? Would the effectiveness of this process be relatively low?
   2. In the second stage, the model uses MLLM evaluations to guide image editing. However, image editing may struggle to handle complex cases, such as spatial collisions between objects.
* The authors claim that image editing serves as a bridge connecting generation and understanding to effectively guide reasoning. I don’t quite understand this claim—image editing seems to act more like an intermediate node between two end-to-end processes, continuously correcting reasoning results rather than guiding a single reasoning process.
* In many cases, unlike the examples involving spatial or numerical relationships, the prompt does not clearly contain an explicit logical relationship, making it difficult to determine correctness. In such cases, it is not straightforward to compare and match the prompt with the generated image for evaluation, which may limit the applicability of the proposed method.
* The method involves an iterative process, but the authors did not provide sufficient discussion about it.
* It is well known that MLLMs are prone to hallucinations. Would alternating between MLLM understanding and generation accumulate hallucination errors and lead to reliability issues in the final outputs? I believe the authors should discuss this or provide corresponding experimental evidence.
* The authors seem to overstate the contribution of their model to optimizing MLLM understanding and generation. The editing process itself is also based on prompt-based image generation. If the MLLM fails to understand the prompt initially, why would it perform better with a different editing prompt? In essence, the method performs iterative optimization, but the understanding and generation processes remain separated.

**Questions:**

See the weaknesses.

---

### Official Review · Reviewer_u2Kd · 2025-10-31

**Soundness:** 2
**Presentation:** 3
**Contribution:** 1
**Rating:** 2
**Confidence:** 5

**Summary:**

This paper proposes Understanding-in-Generation (UiG), a framework designed to enhance the generative capabilities of image synthesis models. By leveraging multi-turn image editing, UiG iteratively refines and corrects undesirable content in generated images. The effectiveness of UiG is evaluated using the TIIF benchmark.

**Strengths:**

* The motivation is well-justified and intuitively demonstrates the logical soundness of the proposed method.
* The paper is clearly written, with coherent and logically structured exposition.

**Weaknesses:**

* The proposed method is overly simplistic and lacks novelty, as it merely combines off-the-shelf image generation and image editing models without introducing substantial innovation.

* The experimental results are questionable: in practice, similar outcomes could be achieved by combining virtually any pair of image generation and editing models, such as FLUX and Kontext, or Qwen-Image and Qwen-Image-Edit, yet the authors do not conduct comparative experiments to substantiate their claims.

**Questions:**

Please refer to the weakness.

---

### Official Review · Reviewer_m4bE · 2025-11-01

**Soundness:** 2
**Presentation:** 3
**Contribution:** 1
**Rating:** 4
**Confidence:** 4

**Summary:**

This paper proposes a novel reasoning framework called Understanding-in-Generation (UiG), designed to enhance the text-to-image generative capability of unified multimodal models. The key insight is to treat image editing as a bridge to infuse the model’s understanding capability into the generative process. At each iteration, UiG first generates an image, then evaluates it using the model’s understanding function to identify misalignments with the prompt, produces an editing instruction, and regenerates the image accordingly. This iterative reasoning allows the model to refine generation through understanding. Experiments on the TIIF and WISE benchmarks demonstrate consistent performance gains over state-of-the-art reasoning methods — a +1.11% improvement for short prompts and +3.92% for long prompts on TIIF, and a +0.16 overall gain on WISE. Qualitative results show superior spatial and semantic alignment in generated images.

**Strengths:**

1. This paper is well-written and is very easy to follow.
2. The idea is very clear and it improves the baseline model.
3. The experiments covers many benchmarks and models, which makes the paper convincing.

**Weaknesses:**

1. The idea is not very novel and lacks technical contribution. Introducing editing into generation process has already been introduced in previous research work: GenArtist: Multimodal LLM as an Agent for Unified Image Generation and Editing (https://arxiv.org/pdf/2407.05600). Although in GenArtist paper, the setting is not exactly the same as this paper, the idea is very similar. It would be great if the author can claim that your idea have clear technical difference and contribution again.

2. The evaluation benchmark is not enough. It would be great to also evaluate on more reasoning-based editing benchmarks which released several month before the ICLR submission deadline: (1) R2I-Bench (https://arxiv.org/pdf/2505.23493); (2) T2I-ReasonBench (https://arxiv.org/abs/2508.17472). The reason to incorporate more benchmarks is that according to the two papers, they covers different types of reasoning capabilities compared to previous benchmarks that this paper conduct evaluation on. Because this paper is about improving reasoning capabilities, it is fair to cover as many reasoning type as possible, otherwise, it is hard to claim the effective of this work.

3. From my understanding, this idea is similar to an agent workflow with self-reflection, which means that we can use different image generation, understanding, and editing model to achieve this workflow. Therefore, I think it is fair to compare Bagel with Bagel+UiG. But comparing Bagel+UiG with other baselines is not fair because they use different baselines. Therefore, I'd like to see that:

 (a) Could authors claim the fairness in the Table 2?

(b) Adding more comparison experiments, such as Janus-pro v.s. Janus-pro + UiG, or other similar base model v.s. base model + UiG.

(c) Try to use different combination of models, such as Flux for generation + QwenVL for self checking + Flux1-kontext for editing.

4. The improvement of UiG is marginal --  only 1.11/3.92 for short/long prompts. It is not very promising due to its very complex pipeline which requiring several times of inference time compared to other baselines.

**Questions:**

1. Could you explain why in some cases in Table 1, UiG performs worse than previous baselines such as for Rela./Reas., UiG performs worse than T2I-R1?

---

### Meta-Review · Area_Chair_oE4w · 2025-12-08

**Summary:**

The reviewers raised numerous concerns about the technical contribution and the evaluation. The concerns include similarity to previous work, combining existing techniques without much novel insight, and the extent of the evaluation to judge the contribution. All four initial reviews were negative, and no rebuttal was provided.

**Reviewer Concerns:**

No rebuttal was provided so all reviewer concerns are still unaddressed.

**Reviewer Scores:**

I predict that all reviewers would have kept their scores because no rebuttal was provided.

---

### Decision · Program_Chairs · 2026-01-26

Reject